# Electrophysiological Remodeling: Cardiac T-Tubules and ß-Adrenoceptors

**DOI:** 10.3390/cells10092456

**Published:** 2021-09-17

**Authors:** Peter T. Wright, Julia Gorelik, Sian E. Harding

**Affiliations:** 1School of Life & Health Sciences, University of Roehampton, Holybourne Avenue, London SW15 4JD, UK; peter.wright@roehampton.ac.uk; 2Cardiac Section, National Heart and Lung Institute (NHLI), Faculty of Medicine, Imperial College London, Hammersmith Campus, Du Cane Road, London W12 0NN, UK; j.gorelik@imperial.ac.uk

**Keywords:** electrophysiology, cardiac, t-tubules, microdomains, arrhythmia, beta-adrenergic, sympathetic nervous system, cAMP

## Abstract

Beta-adrenoceptors (βAR) are often viewed as archetypal G-protein coupled receptors. Over the past fifteen years, investigations in cardiovascular biology have provided remarkable insights into this receptor family. These studies have shifted pharmacological dogma, from one which centralized the receptor to a new focus on structural micro-domains such as caveolae and t-tubules. Important studies have examined, separately, the structural compartmentation of ion channels and βAR. Despite links being assumed, relatively few studies have specifically examined the direct link between structural remodeling and electrical remodeling with a focus on βAR. In this review, we will examine the nature of receptor and ion channel dysfunction on a substrate of cardiomyocyte microdomain remodeling, as well as the likely ramifications for cardiac electrophysiology. We will then discuss the advances in methodologies in this area with a specific focus on super-resolution microscopy, fluorescent imaging, and new approaches involving microdomain specific, polymer-based agonists. The advent of powerful computational modelling approaches has allowed the science to shift from purely empirical work, and may allow future investigations based on prediction. Issues such as the cross-reactivity of receptors and cellular heterogeneity will also be discussed. Finally, we will speculate as to the potential developments within this field over the next ten years.

## 1. Introduction

The genesis of the mechanical pumping of the heart results from the synchronous contraction of individual cardiomyocytes. This activity is externally moderated by many factors such as the extracellular matrix, vasculature, the autonomic nervous system, inflammatory factors, and many more. However, the pathophysiology of arrhythmia ultimately emanates from the change in the activity of these cells. Thankfully, the simplistic view of cardiomyocytes as coming in two variants, atrial or ventricular, seems to be waning [1,2]. The reality is that the four chambers of a healthy heart are replete with sub-domains containing myocytes which, structurally, may bear more resemblance to cells from other chambers than their immediate neighbors [3]. It is therefore impossible to consider arrhythmia and the electrical remodeling of the heart without an analysis of the sub-cellular organization within populations of myocytes. The study of the t-tubules and the wider transverse-axial-tubules system (TATS) has been intrinsically important in the recognition of this structural heterogeneity within the cells of the myocardium [4]. Within atrial populations, there are reported to be multiple structural types containing either ventricular-like cells with a full TATS, cells with some tubulation, and cells in which a TATS is absent [3]. For many years, it was believed that atrial myocytes did not contain a TATS, the rationale being that they were too small to require the effects of voltage activated ion-channels to be conducted into the cell interior [5,6]. This view was overturned by studies in large mammals and rodents with the advent of improved cell isolation and imaging techniques [7,8]. Ventricular cells were hypothesized to be completely tubulated throughout the myocardium. This has proven to be an incomplete picture as experiments on healthy rodents reveal that the cells of the apical left ventricle have a sparser TATS than the cells taken from the basal left ventricular myocardium [9]. This is hypothesized to be due to load differences within the heart, as mechanical unloading disrupts the normal TATS structure of the left ventricle [10]. A more appropriate view of this situation may therefore be that cell size and TATS (although indirectly linked) simply develop to match the mechanical niche of the particular section of the myocardium.

## 2. Electrophysiological Remodeling of the Heart and TATs

There are manifold etiologies which may initiate cardiac electrophysiological remodeling, but all pathophysiological processes can be characterized as either contributing to the trigger or the substrate of arrhythmia [11]. Broadly, these physical processes may be assumed to be physiological and structural, respectively, but practically there is an overlap. The molecular and cellular processes can be simplified to issues which affect the normal ionic fluxes within cells, the ability of cells to integrate, and changes in macro-cellular structure (affecting action potential). The exact contribution of these processes to arrhythmogenic phenomena, such as early and delayed after depolarizations, rotor formation, and ventricular tachycardia trains, is covered more extensively elsewhere [12,13,14].

The remodeling of the TATS tends to be viewed as the passive loss of the normal density and regularity of TATS of cardiac cells. In the situation of cellular hypertrophy this is imagined as the dispersal of the TATS membrane into the external membrane of a growing cardiomyocyte, resulting in the decay of the tubules from a perfect regular system. Several threats to this view have emerged over the past years.

First, there appears to be multiple t-tubular types, many of which are not transverse, and have led to the term TATS becoming more commonplace as axial tubules are understood to be an essential part of the system (in fact, this has been understood for nearly four decades [15]). Axial tubules may, in fact, have different electrophysiological properties as compared to their transverse counterparts with respect to calcium handling [16] at different junctures in pathology, and it is unclear if they are intrinsic to the maintenance of function or push the tissue toward further damage [17]. Within end stage heart failure there appear to be giant wedge formations in the transverse plane of the cardiomyocyte, but it is unclear if this too is a passive effect or if these structures have some physiological relevance [18].

Second, many of the studies of t-tubulation in the context of heart failure come from murine or small animal models. These species are understood to have higher t-tubular density than larger mammals (perhaps indicating the role of the TATS in maintaining beat-to-beat consistency in contraction, where a high heart rate is required) [19]. Even in these models, we observe that there is variation in the density, regularity, and the configuration of the TATS within the cardiomyocytes of healthy rodent hearts [9]. This contributes to the view of the TATS as being significantly plastic, maintaining a dynamic equilibrium, and adapting to the electro-mechanical needs of the heart [20]. The third threat to the view of the TATS as a monolithic network, with a rigid, perfect structure which is lost in pathology, is its interaction with other cellular microdomains. For example, the TAT system has been indicated to be a framework for caveolae and is the site of multiple ‘caveolations’ [21,22]. It is unclear what the physiological relevance of these multiplexed membrane systems is at the current time.

Third, the loss of TATS is intrinsically linked to the altered expression of the scaffolding molecules which stabilize the tubular elements. The panel of molecules involved in this process comprises Junctophillin-2 (JPH2) [23], bridging integrator 1 (BIN-1) [24], caveolin-3 (Cav3), and cholesterol [25]. These molecules are both responsible for maintaining the morphology of the tubule and the electrophysiological characteristics of the structure by anchoring and controlling molecules such as ion channels. Therefore, the situation is potentially complicated by genetic factors and the combined mechanisms of tubular loss, loss of electrophysiological organization, and the unexpected properties of molecules such as JPH-2 which act as a transcription factor [26]. This all occurs against the backdrop of highly specialized, cardiac sub-domains. JPH-2 is understood to control the structure of the calcium dyad by linking ion channel function to control calcium-induced calcium release [27], as well as being involved in the maintenance of normal tubular morphology [28]. BIN-1 is also crucial for the maintenance of ion fluxes in healthy ventricular cardiomyocytes [29]. Caveolae appear to control distinct sub-populations of calcium channels within the cardiomyocyte membrane, with ramifications for electrophysiology [30].

## 3. A Focus on the Ventricular Myocardium

The atria contain some of the most important myocardial sub-domains from the perspective of arrhythmia generation. The sinoatrial node contains specialized myocytes which contain virtually no TATS but are filled with other important signaling microdomains such as caveolae [31]. It is outside of the remit of this review to discuss the interaction of adrenoceptors and TATS in the setting of electrical remodeling in cells which are not normally tubulated. Therefore, this review will essentially discuss the cells of the areas of the myocardium which are normally tubulated, namely the ventricles; the ideas can be extrapolated to the tubulated trabeculae of the atria where empirical data are unavailable. Ventricular cardiomyocytes do not have a pace-making role and, therefore, arrhythmic events emanating from these cells relate to focal behavior (aberrant electrical behavior in small groups of cells) or non-conduction of action potentials. Focal behavior is primarily contributed to by cellular events such as early after depolarizations (EADs) or delayed after depolarizations (DADs). EADs and DADs are caused by the mis-activation of cellular ionic currents; this activation may affect cells in the vicinity and has ramifications for force production, as the cell’s contraction is not synchronized with the rest of the tissue. Electrical desynchrony between cells within the tissue is the primary culprit in the phenomena of non-conduction of action potentials. This can be caused by a lack of junctional connectivity between cardiomyocytes at the gap junctions. Weakly coupled cardiomyocytes cause arrhythmogenic foci in failing hearts [32]. Delays in conduction of APs can also be caused by scar tissue and fibrotic processes which are produced within the heart in response to hypertrophy or to replace cardiomyocytes after cell death. In the following section we describe the nature of beta-adrenergic receptors and the electrophysiological processes they control in ventricular cardiomyocytes.

## 4. The β-Adrenergic Receptors

The sympathetic stimulation of the heart is central to the maintenance of normal organ structure and function, as well as adaptation to the rigors of mammalian life [33]. Within the myocardium, this control is exerted in part by neurohormonal activation of the family of adrenoceptors comprising the alpha and beta sub-families. In this review, we will concentrate purely on the involvement of beta-adrenoceptors in electrophysiological remodeling. However, it is important to remember that alpha-adrenoceptors have a significant role in the control of the myocardial vasculature and, consequently, arrhythmogenesis resulting from cell death or ischemia [34]. Physiological neurohormones will activate both receptor families, therefore there will be an interplay between the effects of both systems.

Beta-adrenoceptors (βAR) are members of the Rhodopsin-like super family of G-protein coupled receptors [35]. There are three known βAR subtypes (β1AR, β2AR, and β3AR). β1AR couples to Gs and exerts positively inotropic effects on cardiac function. β2AR and β3AR are pleiotropic; β2AR may exert effects through either Gs or Gi, whereas β3AR stimulates Gi or eNOS [36,37]. As a result, β2AR and β3AR effects are either cardiostimulatory, inhibitory, or cardioprotective, depending on the effector panel activated. The molecular pathways controlled by βAR subtypes are reviewed elsewhere [38], but broadly fall into three categories: the control of adenylate cyclase, by Gα G-protein subunits and the consequent activation/deactivation of the cAMP dependent effectors; the pathways activated by Gβγ G-protein subunits; or those activated by β-arrestins [38,39]. This complicated picture has recently been enhanced by the finding that β2AR will directly inhibit β1AR activity through highly localized mechanisms [40]. The four canonical cAMP effectors are PKA, exchange protein activated by cAMP (Epac) [41], hyperpolarizing cyclic nucleotide channels (HCN) [42], and popeye domain-containing peptides (POPDC) [43].

## 5. cAMP Restriction and Compartmentation

The concept of organellar compartmentation of biochemical events is highly intuitive to cell biologists, but its true relevance to pharmacologists has only been discussed in the last 2–3 decades. The empirical studies of these phenomena are even more recent. The seminal studies of the localization of cAMP by t-tubules were performed by Viacheslav Nikolaev [44,45,46] and others in the Lohse and Gorelik laboratories, βAR were the model receptors and ventricular myocytes the model system. In these experiments, cAMP appears to be localized by t-tubulation [44]; this is speculated to be achieved by several mechanisms not limited to PDE hydrolysis: PKA binding; physical restriction; and the localization of receptors themselves. Measurement of the kinetics of cAMP by FRET was, and remains, the primary method of examining the receptor effects. In early experiments, compartment level application was achieved through the application of agonists via modified micropipettes; since then, localized cAMP responses have been studied with localized FRET sensors [47], tethered sensor PDE system [48], and size restricted agonist application [49], as well as predictive computational models [50].

Despite the significant technological advances in the past years, the speculated pathological nature of de-tubulation with respect to cardiac physiology is mainly assumed. This is perhaps unsurprising, given that integrated organ level analysis of cardiac electrophysiology is conventional, but studying cAMP physiology is currently almost completely limited to the whole cell level.

The relevance of de-tubulation of cardiomyocytes and dispersal of βAR to electrophysiological remodeling stems from two competing phenomena relating to molecular compartmentation. First, compartmentation results in the concentration of a molecular cascade to a region of space. This proximity allows signaling to be efficacious. The loss of compartmentation in this context means that the system is no longer modifiable by the adrenoceptors. Counter-intuitively, the hydrolytic activity of PDEs can allow the persistence of high concentrations of cAMP within an effector microdomain through selective localization and source-to-sink fluxes [51]. The second and older notion of cAMP compartmentation is related to this, given that it was based on the idea of the ‘broadcast’ of cAMP. This conceives compartmentation as a suppressive system, where compartmentalized cAMP does not hit effectors and is prevented from exerting controls on effectors. In this context, the loss of t-tubular integrity and the compartmentation of cAMP mean that cAMP begins to hit effectors that it should not in domains that were otherwise restricted [44]. The loss of compartmentation in this context means that electrophysiology is modified by the incorrect activation of effector molecules. Practically, these two systems probably coexist, and new experimental technologies are required to properly delineate the real nature of the biological effects of t-tubule loss.

Similarly, caveolar domains, comprising the molecules Cav3 and cholesterol, are specifically important for the compartmentation of β2AR-cAMP [9,52,53] and, at sub-maximal levels, β1AR-cAMP [54]. A number of mechanisms for this have been suggested, including increased PDE4 hydrolysis [9] and enhanced efficiency of protein phosphatase activity acting on the EC-coupling machinery [53]. The exact interplay between these domains and molecules has not been elucidated due to the lack of experimental techniques. However, improved imaging has revealed bona fide caveolae within t-tubules in rabbit cardiomyocytes [22]. Therefore, it can be assumed that there is some synergy between these domains, with respect to the control of TATS. Concurrent changes in the TATS, caveolae, and Cav3 localization are observable in hypertrophic remodeling in a rat model of myocardial infarction [17]. In the following sections, we will describe the ways in which βAR modulate the contributory molecular and cellular processes which control the electrophysiology of the tubulated cells of the heart. This will the allow us to infer the contributions of βAR in the context of microdomain and electrophysiological remodeling, as well as the effects of remodeling on βAR.

## 6. The Molecular Modulation of Cardiac Electrophysiology by β-Adrenergic Receptors

### 6.1. Calcium

Historically, the central effector of βAR was considered to be cAMP-dependent protein kinase A (PKA). The primary effect of βAR-cAMP-PKA activation is on cardiomyocyte calcium handling. These effects are exerted by the β1AR and β2AR-subtypes in Gs mode [55]. L-Type Calcium channels (LTCC) are phosphorylated by PKA [56], which increases the rate of calcium influx.

βAR-cAMP-PKA was implicated in the enhancement of ryanodine receptor (RYR2) activation, increasing the rate of calcium released from SR-Ca2^+^ stores. A PKA-mediated phosphorylation of RYR2 was believed to be the mechanism [57], but doubt has since been cast on this hypothesis [58]. The activation of RYR2 by βAR also occurs through a more indirect mechanism involving Ca2^+^/calmodulin-dependent protein kinase II (CaMKII) [59]. CaMKII seems to only phosphorylate RYR2 in the dyadic cleft; this mechanism involves nNOS [60]. The action of LTCC induces the release of calcium from the RYR2 in a process known as calcium-induced calcium release (CICR). JPH-2 is intrinsically important to the process, as it is involved in the direct link between the LTCC and RYR2 to maintain the process [27]. This sort of complex should be considered similar to the general structural and molecular compartmentation of βAR by membrane microdomains within the TATS.

βAR-cAMP-PKA (both sub-types) also phosphorylates phospholamban (PLB), the inhibitor of the sarcoplasmic reticulum (SR) calcium transporter (SERCA2A) [61]. The deactivation of PLB by phosphorylation enhances the rate of uptake of calcium, abbreviating the contraction of the myocyte and inducing a positively lusitropic effect on the myocardium. CaMKII also phosphorylates PLB [62]. Calcium sparks (aberrant, intermittent sub-cellular calcium events) are considered as elemental components of arrhythmias and are the result of the dyssynchronous opening of RYR2. βAR stimulation will therefore cause an increase in spark activity allowing these events to aggregate and potentially drive arrhythmia [63].

These effects seem particularly important in settings of chronic βAR stimulation and may involve the enhancement of basal Ca2^+^ levels by βAR-PKA activity and other cAMP effectors such as Epac [64]. Non-cAMP mediated mechanisms may involve the action of β-arrestins [65,66], NO [67], or β3AR-eNOS [68]. The overload of calcium is assumed to cause DADs [69]; this is speculated to be due to its effect on the sarcoplasmic reticulum Ca2^+^ stores or a chloride current, but a more multifactorial cause is now suggested [70,71]. High calcium levels at the action potential plateau may be a cause of EADs [72]. 

### 6.2. Potassium

The slow-inward potassium current (IKs) conducted by the channel formed by the molecules KCNQ1/KCNE1 is directly modulated by the action of βAR-Gs-PKA, which phosphorylates the complex and increases its activation [73]. The electrophysiological consequence of this is the shortening of the action potential and enhanced cardiac function. This process requires appropriate scaffolding of the PKA by the A-kinase anchoring protein (AKAP9- Yotiao). The inwardly rectifying potassium currents IKr (hERG) and the IK1 (Kir2, 3 + 6) [74] seem to be minimally affected by βAR stimulation under physiological conditions [75]. The fast transient outward potassium current Ito,f (Kv4.2 and Kv4.3) [76] can be suppressed by βAR, but it is unclear if this effect is relevant in mature cardiomyocytes and in an acute physiological setting [77].

### 6.3. Sodium

The cardiac sodium channel (Nav1.5) is responsible for the initial voltage dependent depolarisation of cardiac tissue [76]. Separate populations of the channel modulate the action potential upon excitation; the majority of sodium channels act in a way which produces a rapid transient Na^+^ current (INaT). In conditions of sustained depolarization, a separate, much smaller population of sodium channels may cause a longer lasting, non-activating depolarization. This is the late sodium current, or INaL. βAR Gs-PKA controls INaL through both direct PKA mediated activation and CaMKII-dependent mechanisms (CaMKII is somewhat of a downstream effector of PKA due to its dependence on calcium) [78]. Intriguingly, these activation events modulate specific parts of the action potential phase, with PKA effects occurring early in the AP plateau whereas CaMKII effects occur later. This is concordant with the CaMKII activity occurring downstream of PKA. Further βAR effects are exerted by βAR-Gs-cAMP mediated activation of Epac and reactive oxygen species (ROS), which is activated by βAR-Gs via NAPDH oxidase 2 (NOX2) [78].

### 6.4. Hyperpolarization Activated, Cyclic Nucleotide (HCN) Gated Channels and the Funny Current

HCN channels must be covered in this section, although they are unlikely to occur in myocytes which are normally tubulated. These channels modify the pace of myocardial contraction from specialized cells within the sino-atrial node [42]. As the name suggests, their activity is intrinsically modulated by cAMP, which is produced following βAR-Gs activated adenylate cyclase activity. Increased levels of cellular cAMP activate the channels enhancing the rate of spontaneous action potentials produced by the SAN cells increasing the heart rate.

### 6.5. Sodium–Cation Exchangers

At least three notable exchanger channels exist in cardiac tissue to balance the concentration of cytosolic levels of different cations by extruding sodium. The activation of these molecules will modify the degree of membrane depolarization and the strength of excitation–contraction coupling. The sodium–calcium exchanger (NCX) transports three sodium ions for each calcium ion into the cell (forward mode–Ca2^+^ efflux) or out (reverse mode–Ca2^+^ influx). The mode of the channel is determined by the concentrations of various ions and the transmembrane voltage conditions [79]. There are phosphorylation sites on the intracellular loop of NCX, but the physiological relevance of the upregulation/activation of NCX channel activity by βAR is controversial [80]. The modulation of NCX by βAR-PKA seems to occur under certain experimental conditions [81], but not others [82,83,84]. The sodium–potassium exchanger (Na/K-ATPase) exports three sodium ions out of the cardiomyocyte, and imports two potassium ions by hydrolysing a molecule of ATP [85]. The Na/K-ATPase is inhibited at baseline by a molecule of phospholemman. βAR-PKA drives the phosphorylation of phospholemman, removing the inhibition and activating Na/K-ATPase. Na/K-ATPase activity is also directly driven by β3AR activity [86]. At baseline, the β1-subunit of the Na/K-ATPase is glutathionylated (driven by ROS), this modification inhibits the channel. Through a complicated mechanism, β3AR drives the deglutathionylation of the pump, enhancing its activity. The sodium–proton exchangers (NHE1) activity is upregulated by the action of β1AR-PKA [87]. Interestingly, the activation of NHE1 also regulates NCX activity due to the accumulation of sodium, modifying its directionality. NCX is, therefore, directly and indirectly modulated by βAR. NHE1 activity drives the NCX into reverse mode enhancing the accumulation of calcium within the cell driving enhanced excitation contraction coupling. All ion channel complexes influenced by βAR, referred to in this section, are illustrated in Figure 1.

### 6.6. Reactive Oxygen Species

Reactive oxygen species (ROS) are crucial cellular signaling molecules as well as contributors to cellular toxicity. NOX2 appears to be activated by βAR stimulation through a Gs-dependent mechanism. This enzyme produces two super-oxide molecules by oxidizing a molecule of NAPH. As discussed, this synergizes with βAR stimulated CaMKII and PKA activity to alter Ina, which contributes to the alteration of the action potential. ROS may also be produced by βAR stimulation through the general upregulation of oxidative phosphorylation, required to meet the cells enhanced energetic demands [88].

## 7. The ‘Structural’ Modulation of Cardiac Electrophysiology by β-Adrenergic Receptors

### 7.1. Cardiomyocyte Adhesion

Cardiomyocytes build a functional syncytium by connecting to each other at the intercalated discs at the cell ends. These linkages produce myofibers which tesselate, and form a helical wrap of fibre layers from the endo- to epicardium, producing the ventricular wall. Intercalated discs are built through the deposition of desmosomes, adherens, and gap junctions. The cohesion of cardiomyocytes is intrinsically important to the production of force by the cells and consequently the contraction of the myocardium. Tight coupling of the cells is also crucial for the effective propagation of the action potential and the maintenance of normal cardiac electrophysiology. Sympathetic signaling has been demonstrated to be ‘adhesiotropic’, promoting the stronger coupling of cardiomyocytes by causing the βAR-Gs-cAMP-PKA mediated phosphorylation of plakoglobin (PG). This modification drives greater desmosomal contact in a desmoglein (DSG2) dependent manner [89,90]. This process also seems to involve ERK [91].

### 7.2. Cardiomyocyte Hypertrophy

Cardiomyocyte hypertrophy is the primary driver of myocardial hypertrophy. Pressure-overload drives concentric hypertrophy of cardiomyocytes where normal cellular geometry is maintained, whereas volume-overload drives eccentric hypertrophy. In eccentric hypertrophy, myocytes increase in length. Concentric hypertrophy leads to the thickening of the ventricular wall, and a reduction in the size of the ventricular cavity, whereas eccentric hypertrophy leads to an increase in the size of the heart and an enlargement of the ventricular chamber. βAR signaling has diverse roles in these processes and this paragraph is a gross simplification of these processes, which are of significant relevance to electrophysiological remodeling. βAR-Gs-AC-cAMP (mainly β1AR) [92] stimulation directly drives cardiomyocyte hypertrophy by activating Epac1 and pro-hypertrophic genetic programs mainly coordinated by CREB [92,93,94,95]. The βAR-Gi-PI3K-AKT-GSK3β and βAR-Gi-Ras-Raf-MEK-ERK axes are driven by the ‘stimulus trafficking’ of β2AR to cause the activation of Gi and potentially β3AR in Gi mode [96,97,98]. These two ‘non-classical’ Gi driven pathways seem to exert their pro-hypertrophic effects by modifying the signaling of GATA4. As well as driving a ‘hyper-physiological’ gene expression program and changing the molecular profile of the cell, the physical increases in cell size will affect the cell capacitance and intrinsically modify the cells electrophysiology. The changes at the cellular level must necessarily translate to changes affecting the whole heart, whereby the propagation of the action potential is altered, and hence the electrophysiology substrate is reconfigured.

### 7.3. Extra-Cellular Matrix Remodelling

As cardiomyocytes increase in size, the extra-cellular matrix adapts to accommodate the cells. βAR activity also drives this mechanism by activating the expression of a panel of matrix metalloproteinases (MMP) in cardiomyocytes and cardiac fibroblasts. It seems that this process is caused by the ROS-dependent activation of c-Jun N-terminal kinase (JNK), which causes the expression of extracellular matrix metalloproteinase inducer (EMMPRIN) in cardiomyocytes, which in turn causes the stimulation of cardiac MMP2 and MMP9 [99]. Very recently, this picture has been complicated by the understanding that MMPs are both pro- and anti-hypertrophic. These phenomena may be due to study in different cellular and pathological contexts. The cardiomyocyte βAR response which drives these processes intrinsically requires the function of cardiac fibroblasts. These processes are reviewed more extensively elsewhere [100]. Fibrosis is also an essential element of the organ’s maintenance of its integrity against cell death and apoptosis. β1AR signaling is pro-apoptotic, whereas β2AR and β3AR signaling are anti-apoptotic. The effect of βAR is, therefore, highly contextual. However, the generally malignant apoptotic and necrotic effects of chronic βAR signaling have been understood for many years [101]. Aberrant sympathetic activation may, therefore, remodel cardiac electrophysiology by producing lesions which act as a barrier to the conduction of the action potential. This will interact with the other mechanisms discussed above.

### 7.4. Mechano-Transduction

The ventricular wall is under consistent stretch, strain, and shear stress for the entire life of the organism. Mechanical and electrical activity are intimately linked through mechanosensitive molecular signaling within the cardiomyocyte. Adrenergic activity seems to be primarily linked to mechano-transduction through RYR2. As already described, βAR can modulate RYR2 activity through upregulation of ROS, NOS, and through CaMKII (also activated by ROS). This has made CaMKII a target of interest for anti-arrhythmic therapy. The eNOS-Akt-PI3K pathway positively modulates EC coupling in T-tubular, while negatively regulating the β-adrenergic response in a different subset of caveolae at the surface sarcolemma [69]. It would seem likely that this effect probably involves β3AR, given that this receptor also seems to be heavily localized to t-tubules [102] [103]. Nitric Oxide appears to be intimately connected to mechano-electric coupling within cardiomyocytes [104]. Unlike eNOS, nNOS can be directly activated by stretch or catecholamines in cardiac myocytes, and local NO-CaMKII by βAR [105]. These signals can produce arrhythmogenic Ca2^+^ waves from RYR2 specifically from the cellular dyad [106]. There are several stretch activated ion channel groups within the cardiomyocyte, one of which is the tension dependent Na^+^/H^+^ exchanger. Under increased ventricular tension and enhanced after-load, these exchangers will increase ventricular inotropy, perhaps in synergy with sympathetic activation [107]. NOX2, which is under βAR regulation, is a further mechano-sensitive link to cardiac electrophysiology. The activation of NOX2 is a crucial link in the production of X-ROS, which acts a principal modulator of calcium spark activity [108]. Modifying electrophysiological experimental paradigms from the classic cardiomyocyte in a solution in a Petri dish may be required to investigate these phenomena more fully. Isolated cardiomyocytes, which are the primary model when studying these phenomena, are not loaded; afterload is considered to be crucial for mechano-transduction. By placing a cell in a hydrogel which can be stiffened, researchers have been able to investigate cellular electrophysiology under conditions or normal afterload. In these experiments, it has been possible to demonstrate that APD prolongs until alternans develops [109]. An illustration of molecular and organellar pathways modulated by βAR is presented in Figure 2. 

## 8. The Relevance of T-Tubules to the Modification of βAR Driven Changes in Cardiac Electrophysiology

As is evident from the passages above, sympathetic stimulation, transduced by the βAR, coordinates a significant number of molecular and cellular processes which alter electrophysiology, to facilitate enhanced excitation–contraction coupling within tubulated cardiomyocytes. As we have already described, the remodeling of t-tubules is germane in heart failure, and probably exists contiguously with electrophysiological remodeling. Nevertheless, an attempt must be made to establish if any of the processes exist ‘upstream’ of others, making them useful therapeutic targets. As described, conceptually, a problem exists when building hypotheses as to the relevance of de-tubulation to the remodeling of βAR control of electrophysiology. The loss of tubules may in some cases lead to ‘gain of function’ or ‘loss of function’ for the βAR. In this scenario, only empirical data will suffice, and experimental modalities must have t-tubular resolution to allow us to draw conclusions as to the likely results of structural remodeling for the influence of βAR on cardiac electrophysiology. In the next section, we explore the physiological methods researchers have used to explore the effect of t-tubular remodeling on βAR activity.

Perhaps the primary link between electrophysiology, t-tubules, and remodeling of cardiac electrophysiology is the loss of the dyadic ‘neighborhood’ of calcium-handling molecules [110]. Orphan extra-dyadic RYR2 channels appear to significantly contribute to the calcium spark rate within cardiomyocytes, and it must be assumed that this is due to decreased control of the RYR2 within molecular complexes [103]. Losing t-tubules therefore increases the population of ‘orphaned’ RYR2s; this situation may be exacerbated by altered βAR localization and activation. A further complicating factor, is the discovery that there are separate populations of LTCCs within ‘healthy’ t-tubulated cardiomyocytes. The extra-tubular LTCCs are directed towards caveolar compartments and link more closely to CaMKII [30]. One would assume that losing t-tubules also shifts the equilibrium with a greater number of LTCCs entering these CaMKII activation domains, with the effect of increased CaMKII activity and enhanced RYR2 and spark function.

## 9. Experimental Techniques to Probe TAT/βAR Effects in Cardiomyocytes with ‘Tubular’ Resolution

### 9.1. Removing T-Tubules from Cardiomyocytes

Perhaps the most straightforward way of investigating the effect of t-tubular remodeling on βAR activity is to remove t-tubules chemically or pathologically. Chemically, t-tubules can be removed by exposing isolated cardiomyocytes to osmotic shock with a solution of formamide [111]. At baseline, this reduces the calcium transient, indicating that LTCC and RYR2 function are affected by the loss of tubules. On this de-tubulated substrate, specific β1AR stimulation enhances the calcium transient to a greater degree. In this study, the relative ability of ‘sarcolemmal’ or non-tubulated membrane is calculated vs. tubular membranes and the t-tubulated membrane is posited to be less effective at raising the calcium transient. Given the role of the t-tubule as a crucial element in the organization of the dyad, this speaks of an inhibitory role for the tubular localization of βAR activity. These data suggest a ‘gain-of-function’ state can be presumed for βARs’ influence on electrophysiology in a de-tubulated state, with a greater contribution of LTCC activity (ICa) to the action potential and potentially a higher level of calcium persisting within the cardiomyocyte at diastole. These conditions would of course have ramifications for molecules such as NCX and CaMKII, which are Ca2^+^ dependent.

Cells from failing hearts are pathophysiologically de-tubulated, but the complement of βARs is selectively modified, with the loss of β1AR and the maintenance of β2AR and β3AR populations [112]. Failing cells are therefore awkward models for exploring the mechanistic relationships between de-tubulation, βAR and electrophysiology. However, they are highly relevant, especially if we consider classical electrophysiological remodeling on a substrate of pump failure or within the aged myocardium. The phenomena described in the first paragraph would most likely be diluted by reduction in the number of β1AR units, and any persisting effect would be opposed by greater β2AR and β3AR activity.

### 9.2. Directly Applying Agonists to the T-Tubules

Despite the intrinsic importance of the micropipette to the study of cardiac electrophysiology, it only attained the ability to resolve t-tubular microdomain physiology when deployed as a sensor in a scanning ion conductance microscope (SICM) [113]. T-tubular openings are generally only 100–200 nm in diameter and cannot be resolved by even the best optical microscopy. Equally, the topology of the cardiomyocyte means that they cannot be visualized to guide patch clamp studies using conventional means. SICM serves this purpose, as it employs the patch-clamping pipette as a sensor probe and builds a surface topography image including the z-plane (3D). This allows the visualization of the t-tubule openings of a pre-defined area of the cardiomyocyte surface. Nikolaev et al. used SICM scans of the surface of healthy and failing cardiomyocytes to guide the application of isoprenaline to t-tubules or non-tubulated sarcolemma. The cells were pre-functionalized with FRET-based cAMP biosensors (SICM/FRET) [44]. Using this approach, the researchers were able to identify that, in healthy cells, β1AR-cAMP responses could be elicited across the cell surface, whereas β2AR-cAMP can only be activated when isoprenaline was applied to the t-tubule. β1AR-cAMP responses permeated the cell, whereas β2AR-cAMP remained confined within the vicinity of the application site. In de-tubulated failing cells, both β1AR-cAMP and β2AR-cAMP responses can be elicited on the sarcolemma, and in the remaining t-tubules, the cAMP responses permeate the entire cell. One would hypothesize that unconfined cAMP signaling might elicit more widespread PKA activation. Given the centrality of PKA to the control of multiple electrophysiological processes, one would again assume that the de-tubulation would lead to a ‘gain-of-function’ on βAR signaling. In a series of follow-up experiments, SICM/FRET, caveolin-3 was shown to be central to the localization of β2AR-cAMP within cardiomyocytes [52]. Pathological de-tubulation can be elicited by atrophy as well as hypertrophy in the context of failure or even changes in myocardial provenance. In these experiments, it was demonstrated that de-tubulation resulted in ‘gain-of-function’ with respect to βAR-cAMP, but calcium channel function was down-regulated in the case of atrophy [114]. The electrophysiological outcome of this scenario in the wider heart is therefore unclear. These isoproterenol application studies can be regarded as an evolution of smart patch, which is also a method with t-tubular resolution. This methodology has been crucial in the endeavor to understand t-tubular ion channel content and the effect of de-tubulation during failure [113,115,116].

### 9.3. Localized Fluorescent Reporters

FRET-based biosensor technology has been at the forefront of the study of cAMP compartmentation by cellular microdomains; fortunately t-tubules and cardiomyocytes have served as model domains and cells in many of these studies. The organellar localization of FRET sensors recently reached its apotheosis with the production of the ‘CUTie’-based reporters produced by the Zaccolo lab [47]. The AKAP79 localized sensor, CUTie-AKAP79, can be regarded as a tubular sensor. Data published in 2017 indicates that t-tubular βAR-cAMP responses are quick but remain confined by the t-tubules in healthy cells. This relative confinement is necessary for the appropriate activation of excitation-contraction. Where cAMP is allowed to unselectively permeate the cellular compartments, excitation–contraction coupling is not as efficient. In the case of failing (hypertrophic) de-tubulated cells, t-tubular βAR responses were preserved, but there was reduction in the permeance of cAMP into the SR compartments and the myofibrils of the cells. These data seemingly conflict with data from cytosolic FRET probes.

More recently, the researchers involved with these studies have, collectively, taken the view that in many of these experiments the adrenergic stimulus has been supra-physiological (inducing artificial supra-physiological cAMP responses). They have also suggested that cAMP compartment has been more exquisitely localized than was ever conceived, and that many FRET sensors have not been localized enough [117]. This leaves the community in the dark as to what the exact nature of de-tubulated cAMP responses would be, and makes it even trickier to hypothesize what the ultimate relevance the loss of t-tubules has for the control of PKA activity, and consequent effects on electrophysiology.

Ultra-specific fluorescent probes have been used to demonstrate the localization of β2AR to t-tubules in healthy cardiomyocytes. This work demonstrates that, in overexpression systems, βAR loses their normal localization and kinetics, becoming more motile within the cell membrane. Intriguingly, this paper also mentions populations of cells in which β1AR are absent, which is consistent with mosaicism with respect to muscarinic receptors as well [118].

### 9.4. Polymer-Based Agonists

Barthé et al. have described experiments utilizing polymer-bound isoprenaline to selectively stimulate non-tubulated βAR. In these experiments, the size exclusion of the agonist from the t-tubule results, predictably, in the right shifting of the isoprenaline dose response curve. Interestingly, experiments from FRET-based biosensors reveal (by deduction) that specific organellar effects of cAMP at the nucleus are exerted by the tubular β2AR. Cytosolic effects of cAMP (specifically on excitation–contraction coupling) are exerted faithfully by PEGylated isoprenaline, i.e., by populations of βAR on the sarcolemma [49].

### 9.5. Measuring Activity within T-Tubules

Experiments with super-resolution STED (stimulated emission depletion) microscopy revealed that axial elements within the t-tubular system are responsible for the control of rapid calcium handling [16]. Interestingly, these structures proliferated in hypertrophic cells, which is speculated to drive faster excitation–contraction coupling and to be compensatory within this disease state. However, this may also contribute to arrhythmogenic behavior. As well as measuring calcium handling, a significant number of techniques based on electron tomography have been pioneered to assess the dynamics of t-tubule remodeling at almost beat-to-beat resolution [119]. These techniques have demonstrated that significant effects are induced by conventional tissue fixation techniques [120]. This has obscured our ability to properly track caveolar interactions [22] and has obscured fascinating and important dynamical phenomena such as solute transfer (which also indicates electrical properties) [121,122] and species differences in t-tubule morphology [123]. The wider use of these methods could improve our understanding of the links between t-tubule remodeling and electrophysiological changes.

Random-access microscopy (RAMP) represents a modification of confocal and fluorescent microscopy. It permits the measurement of calcium handling and action potentials at multiple sub-cellular locations within a cardiomyocyte. Sacconi and collaborators have deployed this technology to demonstrate that, in failing cells, certain t-tubules are detached from the wider network and are not electrically active [124,125]. Equally, some apparently typically attached elements within the TATS do not transmit action potentials normally, and axial tubules seem to act as a redundancy mechanism [124]. Further experimentation using failing cells from animals after preparation with coronary ligation or osmotic disruption revealed that defective electrical activity around t-tubules may contribute to Ca2^+^-based arrhythmogenesis by favoring increased spark activity and asynchronous calcium release. RAMP microscopy reveals ‘failure-event’ action potentials which occur in the vicinity of detached t-tubules. Disorganized calcium time-to-peak and delay in re-uptake occur in this context. Importantly, for this review, this study demonstrated that these changes were not exacerbated by application of isoprenaline [126]. These findings were replicated in a study which demonstrated that βAR stimulation increases the frequency of Ca2^+^ sparks and reduces Ca2^+^ transient variability [127]. However, βAR hastens the decay of Ca2^+^ transients similarly in control and HF. βAR stimulation in HF accelerates Ca2^+^ rise exclusively in the proximity of tubules which conduct the action potential, the delayed Ca2^+^ rise found at t-tubules that fail to conduct the action potential is not affected by beta-adrenergic signaling. This suggests the blunted effect of the β-adrenergic signaling may be directly caused by the lack of electrical activity. Impaired EC-coupling and adrenoceptor responsiveness probably results from failure to initiate LTCC activation. Here, the ‘substrate’ is lost, therefore the function is lost. The rate of Ca2^+^ reuptake increases in all parts of the cell, so it seems that PLB is still phosphorylated by βAR-PKA activity, despite t-tubule loss [127]. Here, we see an example of the potential, for both ‘gain of function’ and ‘loss of function’ in the case of de-tubulation with respect to βAR activity. βAR activation may continue to activate calcium handling and EC-coupling, but be completely impeded in the context of the loss of electrical activity. This resonates with the suggestion that TATS regularity is not completely consistent, and therefore these populations of detached or electrically inactive cells may have a purpose in controlling normal physiology. In cells from HCM mice, very minimal alterations were required to alter calcium handling in otherwise regularly tubulated cells [128]. The EC-coupling power and synchronicity due to de-tubulation and reduced βAR response will affect mechanical efficiency. However, patches of electrically inactive membrane may act as a barrier to arrhythmic activity, preventing it from being propagated elsewhere. Specifically, it may prevent DADs by changing the coupling of RYR2. However, orphan RYR2 are shown to induce aberrant spark activity, and the loss of TATS may itself cause aberrant Ca2^+^ waves due to the loss of synchronicity and alter myofilament activity. This may induce arrythmia through mechanoelectrical feedback [129]. Experimental methods with t-tubular resolution are illustrated in Figure 3. 

### 9.6. Computational Models

The goal of one of the earliest mathematical models of βAR behavior was focused on understanding mechanisms of ionic control which contribute to EADs [72]. Therefore, the literature regarding the study of arrhythmia and βAR is quite well developed and has been reviewed in excellent detail elsewhere [130,131,132]. There are, however, some important issues which should be covered here. These models have mainly focused on ventricular myocytes, but studies which specifically focus on atrial cells (with significant ramifications for electrophysiology) are beginning to appear. The role of t-tubular compartmentation has been incorporated into the models in a couple of studies [133,134]. Beta-adrenergic behavior has been directly investigated in a computational model of heart failure [135]. These computational models have been invaluable in uncovering new electrophysiological processes and explaining emergent phenomena; however, incorporating organellar cAMP/PKA compartmentation remains a significant problem in the area. We would therefore posit that computational modelling represents a significant avenue of further research in this area. More sophisticated models should be able to incorporate the molecular processes more comprehensively, as well as introducing elements of anatomical macrostructure and 4D effects from the entire organ. The action potential duration varies during normal human physiology, this is hypothesized to be due to the activity of oscillations in the βAR stimulation of the tissue. Computational modelling reveals that afterdepolarizations and spontaneous APs were induced by phasic β-AR activity with high levels of Ca2^+^ overload [136]. This work specifically suggested a role for mechano–electric coupling and stretch in the phasic modulation of APD. Understanding the effects of β-adrenergic stimulation and stretch on calcium and force dynamics computational models of the cardiomyocyte remains a popular topic of research [137].

### 9.7. Whole-Heart Models

Studies of single cells are, intrinsically, a means to an end. When considering questions of cardiac electrophysiology, we are ultimately interested in the phenomena occurring within the integrated myocardium in a native heart. Recent improvements in Langendorff, whole-heart optical imaging experiments have allowed researchers to better assess t-tubulation and arrhythmia in intact preparations. The Song lab have implemented highly novel confocal microscopy techniques to demonstrate the de-tubulation of epicardial cardiomyocytes [138]. The Efimov group have pioneered whole-heart imaging with a variety of fluorescent probes, measuring calcium handling and conduction within healthy donor and diseased myocardium [139]. In these experiments, they have discovered that the balance of βAR activity undergoes sub-type specific remodeling. Specifically, in healthy hearts, β1AR has a slightly pro-arrhythmic character, whereas in failing hearts (in which we can assume hypertrophic de-tubulation and receptor de-compartmentation has occurred), β2AR activity becomes belligerent. Further study with a similar model system using rat hearts explored the attenuation of alternans by βAR stimulation in the border zone of a healed infarct [140]. On the other hand, in more experiments with human hearts, βAR stimulation may enhance the dispersion of repolarization transmurally, which may be dangerously arrhythmogenic in certain patients [141]. Dual optical mapping has also been utilized to uncover the role of βAR-CaMKII dependent activation of a population of ‘small conductance’ calcium-activated potassium channels. These studies suggested this to be a pro-arrhythmic mechanism within the hypertrophied heart of rats [142]. Weakly coupled myocytes are implicated in created calcium-dependent arrhythmic foci [32]. As well as imaging electrophysiology, an increased number of studies are appearing where FRET based cAMP-biosensors have been measured in whole-heart ex vivo preparations [9]. One could imagine a powerful future combination of these techniques to explore the ramifications of decompartmentalized βAR signaling within an intact heart and its effects on cardiac electrophysiology. As well as optical imaging, the use of multi-electrode arrays has the potential to be very useful for the investigation of autonomic effects on cardiac electrophysiology [143]. Very recently, this approach has been used to describe a further role for cellular uncoupling in arrhythmic mechanisms in the whole heart [144].

## 10. Future Directions

As should be apparent from the preceding text, the modulation of cardiac electrophysiology by βAR signaling has numerous complexities not limited to the effects of t-tubular (and other) microdomain localization, subtype-specific behavior, cellular location, dynamics, and pathological substrates. βAR, in of itself, is neither pro- nor anti-arrhythmic, but with the loss of micro-domain integrity comes miss-sense within the βAR signal transduction. This leads inexorably toward the worsening of arrhythmia and the loss of pump function. Studies have, necessarily, focused on the interaction of one or two of these variables. Methodologies which can directly resolve electrophysiological behavior at the level of the microdomain in the presence or absence of βAR signaling are required to move the field forward. Ideally, these studies would also be able to interrogate this behavior in the intact organ and in a sub-type selective manner; many studies use isoprenaline, and the results are therefore the integrated physiological effect of all βAR function. ‘Cell-in-Gel’ methodologies may represent a useful medium-term solution to this, by permitting the study of cardiomyocytes under normal after-load [109]. These integrated methods are required alongside enhanced computational modelling, which will permit better hypothesis generation.

The need for progress with these studies is pressing from a translational point of view, as we are beginning to understand that the loss of microdomain integrity is an excellent indicator of prognosis and the likelihood of functional recovery of diseased myocardium [18]. Sophisticated ex vivo experiments which must be regarded as very close to the current gold standard indicate the significant, malignant changes in electrophysiology and in the βAR signaling which are induced in the progression to failure. It is likely that successful management of these changes would represent a positive therapeutic pathway. It is also likely that current mechanical device therapies will not reach their potential as a ‘bridge to recovery’ without correcting these pathologies. Equally, βAR signaling is likely to be intrinsic in the establishment of normal microdomain configurations and electrophysiology within the cardiomyocytes, so these processes must be understood to optimize tissue engineering approaches. The advent of high-throughput cardiomyocyte contraction analysis may also allow the findings made at the level of the molecule and microdomain to be translated more rapidly into an understanding of structural/functional differences between populations of cardiomyocytes. This may be relevant to electrophysiological differences within different myocardial layers and regions [145].

## Figures and Tables

**Figure 1 cells-10-02456-f001:**
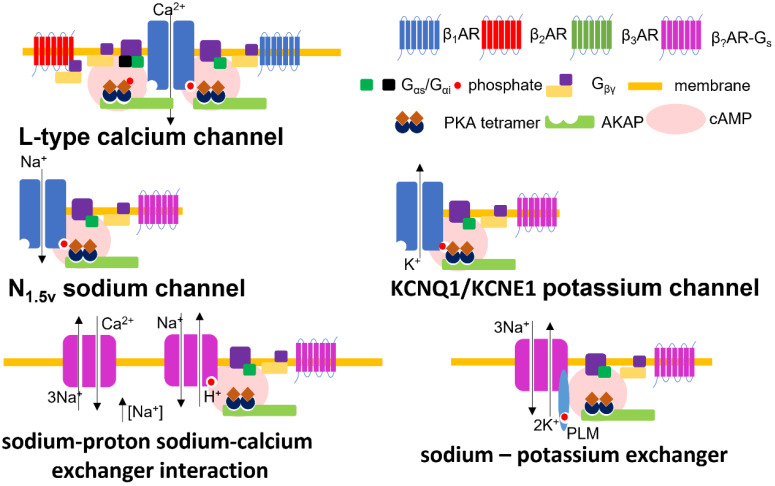
βAR pathways influencing electrophysiology–ion channel membrane complexes. This schematic indicates the suggested molecular complexes between ion channels and βAR subtypes where this is known, along with phosphorylation sites.

**Figure 2 cells-10-02456-f002:**
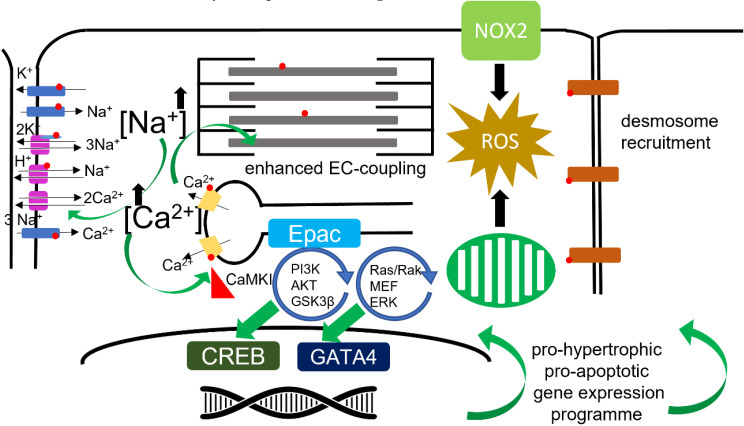
Downstream pathways influencing electrophysiology. This figure represents the pathways downstream of βAR. Kinase pathways and organellar involvement are indicated.

**Figure 3 cells-10-02456-f003:**
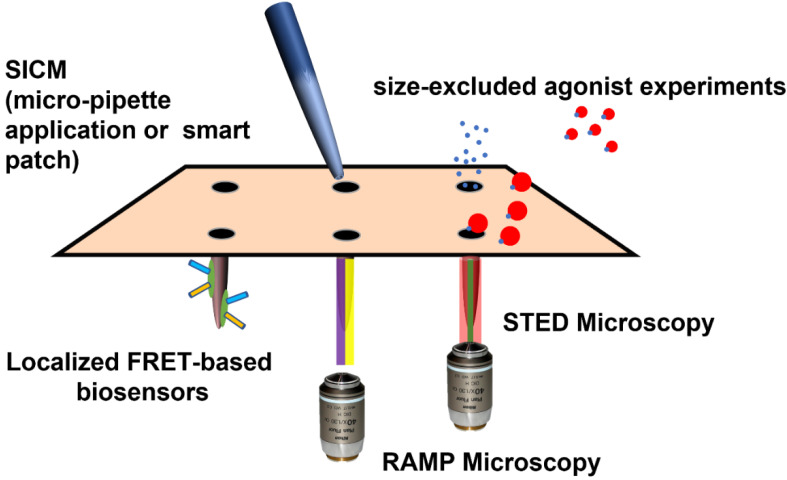
Existing experimental modalities with t-tubular resolution, permitting analysis of βAR physiology.

## Data Availability

Not Applicable.

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
