# Peer review of "Electrophysiological Remodeling: Cardiac T-Tubules and ß-Adrenoceptors"

_cells, 2021, doi:10.3390/cells10092456_

Round 1

Reviewer 1 Report

The manuscript by Wright et al., discuss the intimate connection between cardiac T-tubules and beta-adrenergic response. Overall, the work is nicely focused on ventricular cardiomyocytes but it misses referring to significant literature. The work is acceptable following correction as detailed below.

Major:

The work doesn’t properly cite literature related to t-tubule electrical remodeling that has already answered key questions. In particular, this reviewer is referring to the works from Dr. Sacconi’s group in Florence that has pioneered a technique to probe action potential and calcium with single t-tubule resolution. Major results:

  • PMID 22451916 and 25288764 and 26714042: These works demonstrated that in disease t-tubules can show electrical remodeling without structural remodeling and can generate a delayed local calcium transient, that in turn results in overall calcium delay in disease cells. These works demonstrate the existence of a novel sub-population of t-tubules.
  • PMID 27598150: this work shows that electrically failing t-tubules cannot respond to beta-AR stimulation to accelerate Ca2+ rise (Figure 1), and that RyR-ca2+ sparks are increased with isoproterenol. Demonstrating the unique physiology of these subgroup of t-tubules in relation to beta-AR response.
  • The group has assessed electrical propagation and solute diffusion of TATS in PMID 28507142 (foundational work to citation 109) and has significantly contributed to raise awareness about electrophysiological vs. structural remodeling of t-tubules and their functional interrogation (PMID: 27981580, 30410446, 253685802, 3846763)

In section 8.4 related to mechanotransduction, the authors could comment on the work PMID 34326268 that used a hydrogel 3d encapsulation of cardiomyocytes to study mechanical load and calcium and that has also identified NOS as a crucial player in mechanical signaling.

Visualization of β-adrenergic receptors dynamics in t-tubules has been recently performed with fluorescent labeling in 34088840

Minor:

  • Line 86: the effects of both systems.
  • Line 93: Where are they reviewed? Authors should propose reference.
  • Line 111: is speculated to be achieved
  • Line 160: a parenthesis is opened and never closed

Author Response

Dear Reviewer,

Our sincere thanks for you thorough review and useful comments which have improved this manuscript significantly.

Please find a point-by-point response below

The work doesn’t properly cite literature related to t-tubule electrical remodeling that has already answered key questions. In particular, this reviewer is referring to the works from Dr. Sacconi’s group in Florence that has pioneered a technique to probe action potential and calcium with single t-tubule resolution. Major results:

  • PMID 22451916 and 25288764 and 26714042: These works demonstrated that in disease t-tubules can show electrical remodeling without structural remodeling and can generate a delayed local calcium transient, that in turn results in overall calcium delay in disease cells. These works demonstrate the existence of a novel sub-population of t-tubules.
  • PMID 27598150: this work shows that electrically failing t-tubules cannot respond to beta-AR stimulation to accelerate Ca2+ rise (Figure 1), and that RyR-ca2+ sparks are increased with isoproterenol. Demonstrating the unique physiology of these subgroup of t-tubules in relation to beta-AR response.
  • The group has assessed electrical propagation and solute diffusion of TATS in PMID 28507142 (foundational work to citation 109) and has significantly contributed to raise awareness about electrophysiological vs. structural remodeling of t-tubules and their functional interrogation (PMID: 27981580, 30410446, 253685802, 3846763)

The manuscript has been updated to review these works in detail. Specifically, we have covered this work in the section regrading novel technologies with resolution at the tubular scale. Please see line 554-591. I hope that the reviewer will agree that we have given the requisite level of attention to these ground-breaking works. We hope that they will forgive this oversight, as it is impossible to imagine that this review could have been published without reviewing this technique and studies described. Finally, we have updated figure three to indicate that RAMP is a technique with resolution sufficient to probably discern beta-adrenergic physiology at the level of the tubule. To ensure that the molecular angle is framed, before covering RAMP we have also included passages which cover the molecular stabilization of the t-tubule on lines 92-106 and 194-208.  

In section 8.4 related to mechanotransduction, the authors could comment on the work PMID 34326268 that used a hydrogel 3d encapsulation of cardiomyocytes to study mechanical load and calcium and that has also identified NOS as a crucial player in mechanical signaling.

We thank the reviewer for bringing this to our attention and we have covered this study, accordingly, please see lines 403, 657 where we cover this.

Visualization of β-adrenergic receptors dynamics in t-tubules has been recently performed with fluorescent labeling in 34088840

We thank the reviewer for bringing this to our attention and we have covered this study, accordingly, please see line 524.

Minor:

    Line 86: the effects of both systems.

Corrected now line 140

    Line 93: Where are they reviewed? Authors should propose reference.

Reference added

    Line 111: is speculated to be achieved

Corrected now line 164

    Line 160: a parenthesis is opened and never closed

Corrected now line 81

Reviewer 2 Report

This is a comprehensive review. My only issue is with the flow of the text, I found myself stumbling a lot while reading. My suggestion would to try and streamline the message and improve the 'readability', also by linking the different sub-sections a bit more tightly. Minor comments: some abbreviations not explained, possibly incorrect use of 'electronic remodelling' (should be electrical?) and 'fibrotic plaque' to describe a scar (never seen it described like this before), some punctuation and grammar issues.

Author Response

Dear Reviewer,

Our sincere thanks for your review which has allowed us to improve this manuscript significantly.

Please find a point-by-point response below.

This is a comprehensive review. My only issue is with the flow of the text, I found myself stumbling a lot while reading. My suggestion would be try and streamline the message and improve the 'readability', also by linking the different sub-sections a bit more tightly.

We thank the reviewer for their advice, unfortunately we have not been able to reduce the size of the piece much due to the breadth of the topic and the advice of other reviewers. However, we have considered the readability of the piece significantly.  In our updates version we have altered the structure of the piece so that the reader ‘zoom’ in from the heart at macroscale, to the t-tubules and then a discussion of molecular events. The techniques are then discussed. We have also modified the different sections so that the connections are indicated more explicitly. We hope the reviewer finds the modifications to have improved the piece.

Minor comments: some abbreviations not explained, possibly incorrect use of 'electronic remodelling' (should be electrical?) and 'fibrotic plaque' to describe a scar (never seen it described like this before), some punctuation and grammar issues.

We have altered abbreviations where we found them unexplained, corrected the use of electronic to electrical and removed the term fibrotic plaque as per the reviewer’s advice.